# Trait Mindfulness and Physical Health among Chinese Middle-Older Adults: The Mediating Role of Mental Health

**DOI:** 10.3390/ijerph192316088

**Published:** 2022-12-01

**Authors:** Yuruo Lei, Jiawei Liu, Xinyu Wang, Zhiqi Deng, Qiufeng Gao

**Affiliations:** 1Global Megacity Governance Institute, School of Government, Shenzhen University, Shenzhen 518060, China; 2Department of Sociology, School of Government, Shenzhen University, Shenzhen 518060, China; 3Department of Clinical Medicine, School of Basic Medical Sciences, Shenzhen University, Shenzhen 518060, China

**Keywords:** trait mindfulness, physical health status, mental health, middle-older adults, urban China

## Abstract

Few studies have focused on the specific effects of trait mindfulness on physical health status, especially from a resilient aging perspective. This study examined the mediating role of mental health in the association between trait mindfulness and physical health status among middle-older adults in urban China. The participants included 188 individuals who were recruited from a community senior university and five community senior care centers. The findings reveal: (1) trait mindfulness has a strong effect on two physical health indicators (self-rated physical health and subjective sleep quality); (2) mental health is a significant mediator in the relationship between trait mindfulness and physical health status; and (3) the mediation role of mental health is more evident in the self-rated physical health model (24.15%) than subjective sleep quality (18.10%). This study improves our knowledge of how trait mindfulness can lead to a better physical health in middle-older adults and can lead to the development of social value communication and effective prevention.

## 1. Introduction

Population aging is one of the most serious social problems of the 21st century around the world, especially in China. China’s population of older people accounts for a larger proportion in the world, and the number of people aged 65 and above was 164.49 million in 2019, far exceeding the second-largest nation, India, with 87.15 million [1]. Previous research indicates that older adults are in more fragile health conditions, as the flexibility of biological systems decreases with age and the potential for health problems increases [2]. The physical and mental health condition of older adults in China is not optimistic. Specifically, nearly 57~74% of Chinese middle-older adults have multi-morbidities [3] and 39.85% of older urban adults have depression [4]. Healthy aging and emphasizing older adults’ physical and mental health have become important development strategies in all countries. Under this special social background, several studies have focused on the key factors affecting the mental and physical health of older Chinese adults. Among them, the effect of mindfulness has received increasing attention.

Mindfulness is specified as the skill of purposefully transferring one’s attention to present-moment experiences with non-reactivity, acceptance, and non-judgment [5,6]. As a unique category of consciousness, mindfulness is divided into trait mindfulness and state mindfulness. Trait mindfulness relates to an individual’s disposition of contrariness in the overall level of mindfulness, which is an inherent capacity of the mind [7] and remains relatively stable [8]. State mindfulness is the degree of mindfulness of a person at any given moment [9,10]. State mindfulness can be stressed by interventions such as meditation training, mindfulness-based stress reduction, and mindfulness-based cognitive therapy [5,11].

To date, a lot of empirical research on the associations between older adults’ health outcomes and mindfulness has mainly given attention to the effects of mindfulness-based treatment interventions [12,13] from the state mindfulness perspective [14,15,16]. These pioneering clinical studies have evaluated the effectiveness of mindfulness-based interventions for the health results of older adults [17]. However, the efficacy of trait mindfulness has not been well-explored in these studies.

One study showed that higher trait mindfulness in older adults’ lives was linked with preferable executive functioning and mental health, such as lower perceived stress and depressive symptoms, and higher life satisfaction [18]. For example, trait mindfulness has been proven to increase with age [19] because considerable evidence has shown that the trait mindfulness components, namely present-moment attention, nonjudgment, and acceptance, are all positively connected to age. In particular, flexible goal adjustment and decentering positively correlated with age, which is significant for adults aged 40 and older [20]. Research by Prakash and colleagues [21] examined the relationship between the virtue of the DMN (default-mode network) and trait mindfulness propensity in older adults and further reported a positive connection in older adults between trait mindfulness and health. Summarily, trait mindfulness may give more ideas to infer how older adults’ mindfulness may affect their physical health. Thus, it is necessary to examine the correlation between trait mindfulness and physical health among older adults. Especially among middle-older adults, the relationship between the physical health of people with higher levels of trait mindfulness and the health of people with lower levels needs to be discovered. However, the utility of trait mindfulness on older adults’ physical health remains to be further examined. In particular, few studies have considered the synergistic effects between trait mindfulness and positive or negative mental or physical health outcomes in middle-aged and older adults. In other words, the mechanisms underlying the associations are not clear. Does mental health play a key role in the connection between trait mindfulness and physical health?

Additionally, aging is a gradual process. Many people can retire early at the age of 45 in China and the population of retired people is large. However, few studies have explored the overall health status of this group. This study focuses on middle-older adults in urban China and aims to: (1) reveal the effects of trait mindfulness on middle-older adults’ physical health condition; and (2) evaluate the mediating role of mental health in the link between trait mindfulness and physical health among middle-older adults.

The mindful sustainable aging theory forms the theoretical bedrock for analyzing the association between trait mindfulness, the mental and physical health of middle-older adults, and healthy aging. One study has shown, with the integration of four theories, including activity, disengagement, successful aging, and gero-transcendence theory, mindfulness sustainable aging suggests that mindfulness has the energy to boost healthy aging by improving the defense from negative beliefs about aging and diminished self-imposed ageist limitations [22]. Fiocco and Meisner described conscious sustainable aging as an ingenious way to back the emotional, cognitive, and physical aspects of healthy aging [23]. Recently, Nilsson developed a model that contained four dimensions of mindful sustainable aging as a comprehensive alternative to the above four theories and tried to prove how mindfulness practices support healthy aging in the biological, psychological, social, and spiritual dimensions [24]. From this perspective, mindfulness may be a valuable resource or strategy to improve the overall health status of older adults.

### 1.1. Trait Mindfulness and Physical Health of Middle-Older Adults

Physical health is defined as the condition of the body, with normal status being without disease or serious illness. Other researchers claim that physical health is an active state, the process of sustaining and developing one’s mental functions, biological, and social activity with the longest life expectancy [25].

Many studies have proven that trait mindfulness is closely linked with various positive physical health status outcomes [26,27]. For example, Orme-Johnson and colleagues [26] have supported the hypothesis that trait mindfulness would increase an integrated physiological health state, accompanied by an alert and orderly mental state. The investigation of Lnager [27] suggested that trait mindfulness was beneficial for physical health, and the high-mindfulness group of people lived longer than those in the low-mindfulness group. The possible reasons for this are that trait mindfulness encourages healthy body weight and healthy dietary habits [28], supports a higher quality of sleep, and positively affects self-reported general physical health [29].

Self-reported health and good sleep quality are two key measured indications of physical health status outcomes. Self-related health is a subjective assessment of health status, and it is probably the most viable, inclusive, and conducive measure of health status [30,31]. In the view of Jylhä, self-related health is connected with the biological and physiological state of the individual organism [31]. Hence, self-related health reflects the individuals’ perceptions of their physical health [32]. Individuals with diseases such as cardiovascular disease, mental problems, various stress, and poor quality of interpersonal relationships have been examined to have more negative self-related health in China [33]. Some studies have explored the association between physical health and trait mindfulness from the cardiovascular perspective and weight. Greeson and Chin [34] discussed that higher trait mindfulness was related to some objective indicators of cardiovascular health. Camilleri and colleagues [35] explored the correlation between trait mindfulness and weight, and their results showed that women with higher trait mindfulness had less chance to be overweight, while men with higher trait mindfulness had lower odds of obesity.

Good sleep quality is another indication of measuring physical health status. Sleep disturbances are a common problem today, and poor sleep quality negatively affects cognitive function, mood, memory performance, motor function, and life expectancy in healthy older adults [36,37,38]. Thus, most countries regard improving sleep health as a key goal of healthy aging. Furthermore, Allen and Kiburz [39] found that those with greater trait mindfulness had better sleep quality and greater vitality. In other words, higher mindfulness is correlated with better physical health in general [40]. Murphy and colleagues [29] believe that trait mindfulness could predict the quality of sleep and self-reported overall physical health.

According to the above empirical literature, it is reasonable to assume that trait mindfulness could affect middle-older adults’ physical health directly. This study proposed the first hypothesis:

**H1.** 
*Trait mindfulness may improve acceptance and non-judgment of disease and middle-older adults with higher trait mindfulness would be likely to have better physical health status.*


### 1.2. The Mediating Effects of Mental Health

Attached to the assumption on the effects of trait mindfulness on the physical health of older adults, there is still one crucial question worth further examination, which is whether trait mindfulness could affect physical health directly or indirectly.

Mental health means that an individual has higher positive affect, and lower negative affect, and their evaluation of life is satisfactory [41,42]. Physical health and mental health are closely related. A succinct review shows that mental problems may cause some physical diseases in late adulthood, such as cardiovascular disease, cognitive function, or decline in physical function. In contrast, good mental health, such as a higher sense of purpose in life, resilience, optimism, and internal control means lower odds of macroscopic infarcts, lower risk of cognitive impairment and Alzheimer’s disease [43,44,45], better physical functioning, preferable health status, and lower healthcare utilization [46].

In view of the characteristics of trait mindfulness, it should be significantly associated with mental health. As we know, trait mindfulness is multi-faceted, and self-awareness (observing and describing) and self-regulation (acting consciously, non-reacting, and non-judging) are the key facets. The self-awareness facet of mindfulness indicates the enhancement of perceptive ability, while the self-regulation facet indicates that cognitive control is more flexible [47]. These facets enable mindful individuals to regulate internal and external attention and stimulate and maintain awareness of the current task [48]. On the one hand, mindful individuals are better at evaluating external stimuli, which helps them avoid making immediate judgments about other people’s information or ideas and emotions when they respond unconsciously. On the other hand, mindful individuals also show high levels of self-regulation, which makes them less likely to exhibit subconscious behavior [49]. Overall, higher trait mindfulness appears to be connected to lower levels of daily negative affect symptoms [40,50,51] and less negative affect during experimentally-induced interpersonal conflict [52].

These studies provide constructive evidence that mental health could play a role in the underlying mechanisms and processes supporting trait mindfulness and physical health. Thus, as shown in Figure 1, the study proposed a second hypothesis:

**H2.** 
*Mental health would have a moderating effect between trait mindfulness and physical health among middle-older adults. People with better mental health would have better physical health status.*


### 1.3. The Present Study

Our study first specifies physical health into two aspects (self-rated physical health status and subjective sleep quality), with mental health acting as a mediating role between trait mindfulness and physical health status. Second, the difference between self-rated physical health status and subjective sleep quality in the mediation model of trait mindfulness and physical health status was examined. Together, these two research questions are reflected in the mediation model (see Figure 1). Specifically, how does trait mindfulness contribute to physical health status, and does the mediating role play a difference between the two indicators (self-rated physical health and subjective sleep quality)?

## 2. Method

### 2.1. Participants and Setting

Before the subjects received the questionnaire test, trained graduate students first explained the content, procedure, and confidentiality rules of our study to the subjects. After obtaining informed consent from middle-older adults, a questionnaire survey was conducted. A total of 193 participants were recruited (range = 44–76 years) from a community senior university and five community senior care centers in Shenzhen, China. Five well-trained postgraduates were present in each location (approximately 10–15 people), and a total of 193 middle-older adults (excluding 5 invalid samples) filled out the survey. Survey completion took approximately 20 min. These participants are not clinical samples. A total of 70.70% of the participants were female. Every middle-older adult taking part in this survey volunteered and all of them signed the informed consent form. Our research was approved by Shenzhen University’s Ethics Committee.

### 2.2. Measurements

#### 2.2.1. Trait Mindfulness

The Chinese version of the Mindfulness Measure [53] consists of 10 items (i.e., “It seems I am ‘running on automatic’ without much awareness of what I am doing”), measured by a five-point Likert scale (1 = never, 5 = always). In the scale, each item was reversed with the total scores ranging from 1 to 50. Higher scores indicate higher mindfulness. In the current study, the Cronbach’s alpha of this scale was 0.787.

#### 2.2.2. Physical Health Status

##### Self-Rated Physical Health Status

The study used one single item by Vingilis and Wade [54] to measure self-rated physical health status. A five-point Likert scale (1 = very good, 5 = very bad) was used to measure one item of self-rated physical health status (e.g., “In my opinion, my general health status is…”). In the scale, the item was reversed with total scores ranging from 1 to 5. Higher scores refer to a higher level of physical health status. Self-related health is a subjective assessment of health status and has been widely adopted in large-scale surveys, especially as a well-established predictor indicator of mortality [30].

##### Subjective Sleep Quality

The study used one single item by Hick [55] to measure subjective sleep quality. A five-point Likert scale (1 = very satisfactory, 5 = deeply discontent) was used to measure one item of subjective sleep quality (e.g., “In my opinion, my general sleep status is…”). In the scale, the item was reversed with the total scores ranging from 1 to 5. Higher scores refer to a higher level of sleep quality.

#### 2.2.3. Mental Health

Mental health was measured with a 5-item list from the China Health and Retirement Longitudinal Study Wave 4 (2018) Questionnaire [56] (e.g., “I can keep positive no matter what happens”; “I feel as happy as when I was young”). Responses were based on a 5-point rating scale that ranged from “almost never” (value = 1) to “almost always (value = 5)”. The score of the items in the last three questions was reversed and higher scores indicate higher mental health levels. The Cronbach’s alpha for the scale is 0.612 in this study.

### 2.3. Data Analysis

Before analyzing the data, missing data were handled by mean imputation in SPSS 24. Standard statistical methods were used to test the hypotheses in the study. Meanwhile, we examined the mediating effect of mental health on the relationship between daily mindfulness and physical health in two steps.

According to Baron and Kenny [57], we first established mediation by carrying a series of Pearson correlations and regression analysis through four steps: (1) examined the direct effect of trait mindfulness on physical health status (self-rated physical health and subjective sleep quality); (2) examined the direct effect of trait mindfulness on mental health; (3) examined the direct effect of mental health on physical health status; (4) examined the effect of trait mindfulness on physical health status controlling for mental health. If all of the four steps are met, then variable Mental health completely mediates the trait mindfulness-physical health relationship, and if the first three steps are met but Step 4 is not, then partial mediation is indicated. The second and third approaches, the Sobel test ($Z=\mathrm{ab}/\surd b^{2}S_{a}^{2}+a^{2}S_{b}^{2}$), and the bootstrap resampling procedure were also used to test the mediation effect.

Second, Model 4 of the SPSS macro PROGRESS [58] was used to analyze the mediation model. A bootstrap sampling procedure was conducted to assess the size of the indirect effect and confidence intervals (CI).

## 3. Results

### 3.1. Descriptive Statistics and Correlation

Table 1 provides descriptive statistics and correlations of trait mindfulness, mental health, self-rated sleep quality, and physical health. The outcomes of the correlation test indicate that trait mindfulness and mental health are positively associated with each other. Trait mindfulness and mental health are both positively associated with self-rated physical health or sleep quality.

### 3.2. The Effects of Trait Mindfulness on the Physical Health of Middle-Older Adults

It has been shown in Table 2 and Table 3 that trait mindfulness (X) has significantly influenced physical health (Y) (for self-rated physical health, *β* = 0.236, *t* = 3.308, *p* < 0.01; for subjective sleep quality, *β* = 0.274, *t* = 3.889 *p* < 0.001). These results confirm our first hypothesis that higher trait mindfulness may lead to better physical health status.

### 3.3. The Mediating Effects of Mental Health on Physical Health

In order to reveal the link between trait mindfulness, physical health, and mental health, this study used bivariate correlation analysis. Trait mindfulness was significantly and positively correlated with mental health, *r* = 0.226, *p* < 0.01, and physical health (for self-rated physical health, *r* = 0.236, *p* < 0.01; for subjective sleep quality, *r* = 0.274, *p* < 0.001). Mental health was also significantly and positively related with self-rated physical health, *r* = 0.292, *p* < 0.001, and subjective sleep quality, *r* = 0.239, *p* < 0.01. The results indicated that the relation satisfied the first two conditions of the mediation analysis.

In Baron and Kenny’s study [57], as well as Chen and Yan’s study [59], hierarchical regression was performed in two mediation analyses to evaluate whether mental health problems play a mediator role between trait mindfulness and physical health. The independent variable in these two mediation models was trait mindfulness (X), the mediating variable was mental health (M), and the dependent variables were self-rated physical health (Y_1_) and subjective sleep quality (Y_2_), separately. As shown in Table 1, gender, age, and education are weakly correlated with trait mindfulness. Thus, these were not selected as control variables in the study.

#### 3.3.1. The Mediating Effects of Mental Health on Self-Rated Health Status

In the self-rated physical health status model, as shown in Table 2, it was discovered in the first regression analysis that a 5.6% variance in self-rated physical health status is accounted for by trait mindfulness, and the model indicates a good fit, *F* (1, 186) = 10.943, *p* < 0.01. Trait mindfulness was a key predictor of self-rated physical health, *β* = 0.226, *t* = 3.158, *p* < 0.01.

The results of the second regression test reveal that a 5.1% variance in mental health could be accounted for by trait mindfulness, and the model shows a good fit, *F* (1, 186) = 9.975, *p* < 0.01. The level of trait mindfulness markedly predicted mental health, *β* = 0.226, *t* = 3.158, *p* < 0.01. The third analysis can be divided into two steps. In the first step, mental health significantly predicted self-rated physical health status, *β* = 0.251, *t* = 4.160, *p* < 0.001. As for the second step, when the influence of mental health on self-rated physical health status was controlled, trait mindfulness had no significant effect on self-rated physical health status, *β* = 0.179, *t* = 2.522, *p* < 0.05.

As is presented in Figure 2, the influence of trait mindfulness on self-rated physical health, trait mindfulness on mental health, mental health on self-rated physical health status, and trait mindfulness on self-rated physical health status controlling for mental health is 0.236, 0.226, 0.251, and 0.179, respectively. Furthermore, the results of the bootstrap resampling procedure using 5000 samples, Standardized indirect effect = 0.092, *SE* = 0.116, <0.01, 95% bias-corrected bootstrap CI [0.1548,0.6123], indicated that mental health is a partial mediator between trait mindfulness and self-rated physical health status.

#### 3.3.2. The Mediating Effects of Mental Health on Subjective Sleep Quality

In the subjective sleep quality model (seen in Table 3), the first regression analysis revealed that trait mindfulness was a key predictor of subjective sleep quality, *β* = 0.274, *t* = 3.889, *p* < 0.001. The second regression analysis revealed that the trait mindfulness level significantly predicted mental health, which was the same as the result in the self-rated physical health model. The first route of these variables found that mental health significantly indicated subjective sleep quality, *β* = 0.186, *t* = 3.354, *p* < 0.01; in the second route, with mental health under control, trait mindfulness had no significant effect on subjective sleep quality, *β* = 0.232, *t* = 3.258, *p* < 0.01.

As shown in Figure 3, the effects of trait mindfulness on subjective sleep quality, trait mindfulness on mental health, mental health on subjective sleep quality, and trait mindfulness on subjective sleep quality controlling for mental health are 0.274, 0.226, 0.186, and 0.232, respectively. Furthermore, the results of the bootstrap resampling procedure using 5000 samples, Standardized indirect effect = 0.066, *SE* = 0.112, *p* < 0.01, 95% bias-corrected bootstrap CI [0.0001,0.2138] also confirm that mental health is a partial mediator between trait mindfulness and subjective sleep quality.

## 4. Discussion

This study predicted the prevalence of trait mindfulness among Chinese middle-older adults, evaluated the effect of trait mindfulness on physical health, and examined the mediating role of mental health in the link between trait mindfulness and physical health.

### 4.1. Trait Mindfulness Levels and Physical Health

First, the current study suggests that trait mindfulness has a strong effect on two physical health indicators (self-rated physical health and subjective sleep quality), which is consistent with the first hypothesis and the mindful sustainable aging theory [22]. The theory is a highly comprehensive and positive approach, emphasizing that older adults with high mindful levels are easy to accept and better prepare for the inevitable decline and death. In the physical dimension, they focus on their body’s present state of being as well as routine physical activities. The effects of the trait mindfulness on subjective sleep quality and self-rated physical health status are 0.274 and 0.236 respectively. The effects of trait mindfulness on subjective sleep quality tend to be stronger than on self-rated physical health status. On the one hand, the above results indicate that higher mindfulness is more connected with good sleep quality than with self-reported physical health status among Chinese middle-older adults. On the other hand, trait mindfulness could be associated with better subjective sleep quality because people with higher trait mindfulness levels show lower negative self-assessment and unawareness, which is consistent with previous studies [60,61]. Meanwhile, self-reported physical health status is a comprehensive index that is influenced by other factors apart from trait mindfulness. Conversely, subjective sleep quality is one single indicator, which is only influenced by trait mindfulness.

### 4.2. Mediating Effects of Mental Health

As assumed, the results show that mental health plays a mediating role between trait mindfulness and two physical health indicators (self-rated physical health and subjective sleep quality) respectively. This result confirms the second hypothesis and is consistent with the results of previous trait mindfulness studies [42,52,62,63]. The characteristics of trait mindfulness, self-awareness, and self-regulation not only help middle-older adults avoid immediate judgments about other people’s information or ideas and emotions, but also make them less likely to exhibit subconscious behavior. Therefore, they are more adaptive to the current state. At the same time, based on the mindfulness sustainable aging theory, we concluded that it would be easier for a middle-older adult with higher trait mindfulness to strengthen resilience to negative convictions on aging, and to minimize self-imposed negative assessment, so that they will be less likely to develop clinical diseases such as depression, and their mental health can be further improved. High mental health levels lead to good physical health (i.e., fewer physical diseases and improved sleep quality). In the present study, mental health is a partial mediator between trait mindfulness and self-rated physical health, and subjective sleep quality. The mediating effects of trait mindfulness on the self-rated physical health status model tend to be stronger than those on the subjective sleep quality model. Specifically, the mental health mediation proportion was 24.15% in the self-rated physical health model and 18.1% in the subjective sleep quality model, respectively. This may indicate that meditation’s role in mental health is mainly to influence middle-older adults’ health status through self-rated health.

### 4.3. Limitation and Future Directions

There are two limitations in this study. First, we used a single item to test sleep quality, although the purpose of this action was to reduce the resistance of the elderly to complete the questionnaire. Second, although the theoretical foundation of previous research has provided insight into the current study and has deepened understanding of it, the cross-sectional design constrained our ability to infer the causal relationships among the variables. Therefore, the results of this study can only indicate the association. Third, although self-reporting was appropriate in view of our study objectives, the validity of the self-rated health status was questioned. In this case, people with the same health level may evaluate their health status in different ways, and there exists heterogeneity in the report, which can probably lead to some bias, such as social desirability bias or other types of bias [64]. Future research should include a larger number of participants and specify the effect of trait mindfulness on sleep quality in the mediating model among middle-older adults in Chinese urban cities.

Nevertheless, our work also presents various strengths. First, the study is helpful for middle-adults to accept uncertainty positively in life which can improve their lives in the city. Furthermore, to the best of our knowledge, this is the first study to explore two indicators of physical health status in the mediating model.

## 5. Conclusions

Our results are of theoretical significance. The research on the process of mindfulness affecting physical health among middle-older adults is still in its infancy. Furthermore, previous studies have relied greatly on mindfulness-based intervention [64,65], but studies have made progress in the positive effects of trait mindfulness on physical health from the perspective of resilience aging and socioemotional selection. In the setting of healthy aging, this study is the first to combine trait mindfulness and physical health status, with mental health as a mediator, in the context of healthy aging. The results suggest that trait mindfulness has both direct and indirect effects on physical health among middle-older adults. Thus, a new theoretical framework can be tested to further integrate established theories of healthy aging with the physical health of older adults from trait mindfulness.

One promising direction is that this study uses the resilience aging theory and self-regulation theory of trait mindfulness. Mindfulness is considered beneficial because it improves self-regulation [40,49], which enables older adults to be resilient and quickly adapt to present moments and situations. Self-related health and sleep quality have been consistently linked to mental health and physical health status. Moreover, trait mindfulness is crucial to mood regulation and sleep quality. The results of the present study help to demonstrate that such efforts may be fruitful.

On the practical implication, there are two contributions in social values and an individual’s life. First, free mindfulness courses can be offered at activity places for middle-older adults to help them master mindfulness skills. Second, mindfulness could help middle-older adults cope with illness and face the uncertainty of life positively, and better adapt to life.

## Figures and Tables

**Figure 1 ijerph-19-16088-f001:**
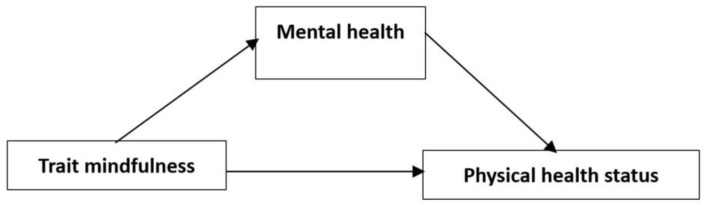
The mediation model of trait mindfulness and physical health.

**Figure 2 ijerph-19-16088-f002:**
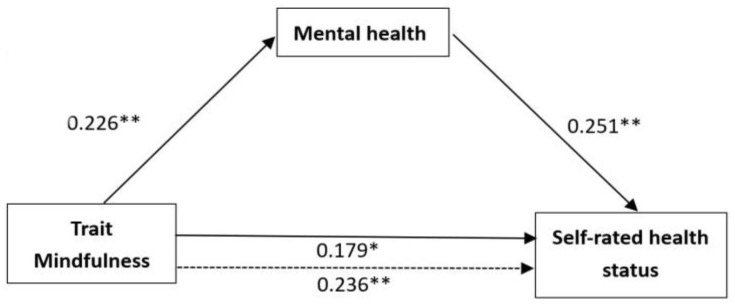
The mediation model of self-rated health (Note: the dotted line denotes that the effect of trait mindfulness on self-rated health is not included as a mediator). * *p* < 0.05, ** *p* < 0.01.

**Figure 3 ijerph-19-16088-f003:**
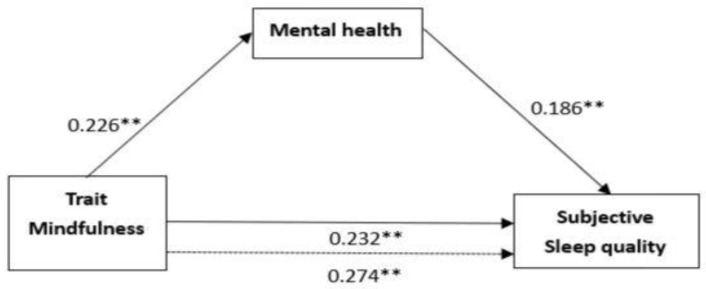
The mediation model of subjective sleep quality (Note: the dotted line denotes that the effect of trait mindfulness on self-rated health is not included as a mediator). ** *p* < 0.01.

**Table 1 ijerph-19-16088-t001:** Descriptive statistics and correlations of study variables.

Variable	*M*	*SD*	1	2	3	4	5	6	7
1. Trait mindfulness	3.631	0.598	-						
2. Mental health	3.881	0.645	0.226 **	-					
3. Self-rated physical health	3.707	0.973	0.236 **	0.292 ***	-				
4. Subjective sleep quality	3.333	0.946	0.274 ***	0.239 **	0.341 ***	-			
5. Gender	1.710	0.456	−0.022	0.193 **	0.252 **	−0.058	-		
6. Age	62.610	6.280	0.027	−0.078	−0.165 *	0.017	−0.281 **	-	
7. Education	4.300	1.755	0.080	0.058	0.043	0.030	−0.129	−0.033	-

Note: *N* = 188. *M* = Mean. *SD* = Standard deviations. * *p* < 0.05, ** *p* < 0.01, *** *p* < 0.001.

**Table 2 ijerph-19-16088-t002:** Results of hierarchical regression analysis of the self-rated physical health model.

	Self-Rated Physical Health	β ^a^	R^2^	Adjust R^2^	F	ΔR^2^	ΔF
	B (SE)						
Analysis One: Trait mindfulness to Self-rated physical health (path c)	0.384 (0.116)	0.236 **	0.056	0.05	10.943 **	0.056	10.943 **
Analysis Two: Trait mindfulness to Mental health (path a)	0.244 (0.077)	0.226 **	0.051	0.046	9.975 **	0.051	9.975 **
Analysis Three: Step 1: Mental health to Self-rated physical health (path b)	0.379 (0.107)	0.251 ***	0.085	0.08	17.304 ***	0.085	17.304 ***
Step 2: Trait mindfulness to Self-rated physical health (path c)	0.291 (0.115)	0.179 *	0.116	0.106	12.082 ***	0.060	12.592 **

Note: * *p* < 0.05, ** *p* < 0.01, *** *p* < 0.001. ^a^ standardized regression coefficients at each step.

**Table 3 ijerph-19-16088-t003:** Results of hierarchical regression analysis of the subjective sleep quality model.

	Subjective Sleep Quality	β ^a^	R^2^	Adjust R^2^	F	ΔR^2^	ΔF
	B (SE)						
Analysis One: Trait mindfulness to Subjective sleep quality (path c)	0.434 (0.112)	0.274 ***	0.075	0.07	15.125 ***	0.075	15.125 ***
Analysis Two: Trait mindfulness to Mental health (path a)	0.244 (0.077)	0.226 **	0.051	0.046	9.975 **	0.051	9.975 **
Analysis Three: Step 1: Mental health to Subjective sleep quality (path b)	0.273 (0.104)	0.186 **	0.057	0.052	11.251 **	0.057	11.251 **
Step2: Trait mindfulness to Subjective sleep quality (path c)	0.367 (0.113)	0.232 **	0.108	0.099	11.222 ***	0.033	6.844 *

Note: * *p* < 0.05, ** *p* < 0.01, *** *p* < 0.001. ^a^ standardized regression coefficients at each step.

## Data Availability

The data presented in this study are available on request from the corresponding author. The data are not publicly available due to some of the subjects’ objection to public data.

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
