# Peer review of "Trait Mindfulness and Physical Health among Chinese Middle-Older Adults: The Mediating Role of Mental Health"

_ijerph, 2022, doi:10.3390/ijerph192316088_

Round 1

Reviewer 1 Report

This study examined previously underreported relationship between trait mindfulness and physical health status among middle-older adults in urban China and to what extent mental health mediates this relationship. Overall the manuscript is well written. The introduction is thorough and provides sufficient justification for the gap in the literature that drives interest in carrying out this study and the relevance of it results. The model, measures used, and final path weights are well explained and clearly displayed. The discussion is on point, making relevant and cogent interpretation of the findings. Major concerns are with the lack of detail regarding ethical approval and analysis used. First, the institutional review board statement states not applicable, but it is unclear why there is no information in the participants section regarding whether this study received approval given an exempt status and if the community centers mentioned gave approval in writing. Second, the analysis mentions Baron and Kenny but does not specify what statistical path analysis modelling tool within SPSS was used, for example the process macro by Hayes. 

Author Response

More details is in the following files

Reviewer 2 Report

1. I suggest organising the introduction differently. I think it would be better to delete the section "Literature review and hypotheses". I would leave a single section "Introduction" and use the sub-sections that section 2 already has.
I think it would be clearer if the two hypotheses were unified at the end of the Introduction in a subsection that could be called "present study" in which the aim and objective of the study and the two hypotheses are described.

2. In the participants' section, I am missing information on consent information, the way in which this was done, the permissions requested from the relevant ethics committee for the processing of individuals' data, etc. I suggest the creation of a sub-section procedure to explain the procedures carried out for data collection. 

3. The Data Analysis section is very small. I miss some more explanation of all the analyses that will be carried out later. Correlations, regressions, moderations. An introductory paragraph of the analysis plan carried out. 

4. The study has other limitations that have been overlooked and which I think it is important to note in the limitations paragraph of the study.

The sample is small and highly variable in terms of age. The study refers to middle-aged adults and the participants are between 44 and 76 years of age. The variability in the time of life is very large and it is problematic to include people of such a wide age range in a single category.
An age cohort design could have been considered and a comparison could have been made.
It is true that being a small sample is the main handicap when it comes to making this type of design by age groups.
This aspect should be developed in the limitations section or considered in the analyses.
On the other hand, I find another major limitation in the data collection instrument: some of the questionnaires used include only one item, which can be a problem when it comes to reliably analysing the construct to be analysed. It is a short questionnaire, and that helps in data collection, but too much brevity can put the reliability and validity of the instruments used to measure a particular construct into problems. This issue should also be addressed in limitations.

5. Finally, after the section on limitations, I would end the manuscript with a concluding paragraph summarising the most important aspects of the study and its practical implications. 

Author Response

More details is in the following file.

Reviewer 3 Report

Dear editors, dear authors.

Thanks for the interesting research.

The topic of the article is very important and relevant.

I have read the entire article with great interest. My comments are aimed at possible improvement of the article. They are not fundamental.

Annotation. It seems to me that the abstract does not need to talk about previous studies, it is better to present the current study in more detail.

Briefly present the results and conclusions. Now they are mentioned, but not disclosed. From the abstract it is impossible to understand what was obtained in the study.

Introduction.

The author cites data that with age (with calendar age) awareness of traits increases.

It is not entirely clear whether this is good or bad in terms of health.

Further, the author cites evidence that mindfulness is positively associated with health in older people.

However, as you know, as people age, their health deteriorates. How can you explain that mindfulness is improving and health is deteriorating?

The review of the literature is interesting, full-fledged, gives an excellent picture of the state of the research of the problem.

Hypothesis H1.

H1. Trait mindfulness can positively impact physical health through wellness exercises or the trait itself, and middle-aged and older adults with higher trait mindfulness are likely to have better physical health.

I would like to clarify. Is exercise still aimed at improving health or at developing the trait itself?

Author Response

More details is in the following file.
